# Immunosenescence, Inflammaging, and Lung Senescence in Asthma in the Elderly

**DOI:** 10.3390/biom12101456

**Published:** 2022-10-11

**Authors:** Tomoyuki Soma, Makoto Nagata

**Affiliations:** 1Department of Respiratory Medicine, Saitama Medical University, Saitama 350-0495, Japan; 2Allergy Center, Saitama Medical University, Saitama 350-0495, Japan; 3Preventive Medicine Research Center, Saitama Medical University Hospital, Saitama 350-0495, Japan

**Keywords:** immunosenescence, senescence, inflammaging, elderly, asthma, neutrophil, eosinophil

## Abstract

Prevalence of asthma in older adults is growing along with increasing global life expectancy. Due to poor clinical consequences such as high mortality, advancement in understanding the pathophysiology of asthma in older patients has been sought to provide prompt treatment for them. Age-related alterations of functions in the immune system and lung parenchyma occur throughout life. Alterations with advancing age are promoted by various stimuli, including pathobionts, fungi, viruses, pollutants, and damage-associated molecular patterns derived from impaired cells, abandoned cell debris, and senescent cells. Age-related changes in the innate and adaptive immune response, termed immunosenescence, includes impairment of phagocytosis and antigen presentation, enhancement of proinflammatory mediator generation, and production of senescence-associated secretory phenotype. Immnunosenescence could promote inflammaging (chronic low-grade inflammation) and contribute to late-onset adult asthma and asthma in the elderly, along with age-related pulmonary disease, such as chronic obstructive pulmonary disease and pulmonary fibrosis, due to lung parenchyma senescence. Aged patients with asthma exhibit local and systemic type 2 and non-type 2 inflammation, associated with clinical manifestations. Here, we discuss immunosenescence’s contribution to the immune response and the combination of type 2 inflammation and inflammaging in asthma in the elderly and present an overview of age-related features in the immune system and lung structure.

## 1. Introduction

Along with the global increase in life expectancy, the percentage of individuals aged > 65 years has increased worldwide. Since an increase in the population of older adults influences asthma prevalence, rate of asthma morbidity among older adults is likely to increase [1,2] as the prevalence of asthma increases globally [2,3]. Moreover, the rate of asthma mortality was highest among older adults compared to other age groups [4,5,6,7] but was lower in childhood and younger adulthood [2]. Recent cluster analyses on asthma in the elderly have generated practical evidence to clarify this issue and to identify specific phenotypes [8,9].

Although asthma in older adults shows unique features, it also has heterogeneous aspects, depending on the disease onset. Asthma can develop in childhood, adolescence, or young adulthood and even after the age of 40, termed late-onset asthma. Cluster analysis on the diversity of severe asthma has helped in understanding the differences of their phenotypes, showing distinct factors associated with type 2 and non-type 2 immune responses that change predominantly with aging [10,11,12]. Studies on type 2 inflammation in asthma have shown that the predominance of atopic factors in childhood asthma translates to the predominance of clinical factors related to eosinophilic inflammation in late-onset adult asthma [12,13]. Evidence suggests a close association between the pathophysiology of asthma and aging. However, the differences in the underlying pathophysiology and clinical characteristics between asthma in older adults and younger adults remain not fully understood.

Human aging results in the comprehensive decline of body integrity throughout the lifespan, due to molecular and cellular aberrations including genomic instability, telomere disintegration, epigenetic alterations, proteostasis defects, dysregulated nutrient sensing, mitochondrial dysfunction, cellular senescence, stem cell exhaustion, and altered intercellular communication [14,15,16]. These irreversible senescent aspects can be recognized as cell dysfunction, impaired response to damage, and alteration in the cellular environment. While prompt senescent changes result in repairable effects on damaged tissue, extended deteriorations with aging are cross-linked and are thought to contribute to age-related chronic disease [17,18,19,20]. Denaturation of intercellular communication between cell and tissue senescence leads to the deterioration of immune reactivation and progression of the inflammatory status, named immunosenescence and inflammaging (Figure 1) [19]. As asthma can occur at all stages of life and remain persistent through the lifespan, it is conceivable that age-related modification of the immune system and inflammation can affect the pathophysiology and clinical manifestation of asthma in the elderly [1].

This article reviews the current understanding of the immunological modifications that occur with aging and their impact on the clinical manifestations of asthma in the elderly and describes immunosenescence and inflammaging.

## 2. Alteration in the Immune System through the Lifespan

The immune system changes its functionality with aging. This age-related change shows a gradual deterioration of the immune response against numerous antigenic or intrinsic sterile stimulations, including excessive elevations, inadequate decreases, and dysregulated immune reactions [17,18,19,20,21,22,23]. Age-related deterioration of the immune system affects both innate and adaptive immunity, leading to advanced susceptibility to and more severe consequences of infection and progression of autoimmune diseases and age-related diseases [20,24]. Incremental repetition of pathogen-associated molecular patterns (PAMPs) and damage-associated molecular pattern (DAMP) stimulation continuously move immune cells from the quiescent to the activated state and enhances tissue damage, resulting in a continuous immune cell cycle and exhaustion, which, in turn, causes cell senescence and increases the production of proinflammatory mediators and DAMPs. In aging, such a noxious loop leads to the progression of the dysregulated immune responses. Immunosenescence has been mainly considered as a decline of the adaptive immune response, including a transition of elevated proportions of memory B cells and T cells from the population of naïve cells, decreased variety of antigen receptor, and changes in the intercellular condition [25,26,27,28,29,30,31,32,33]. Further, recent studies have shown the involvement of the innate immune pathway, particularly regarding phagocytes, in immunosenescence [34,35].

## 3. Innate Immune System Senescence Paradoxically Promotes Chronic Inflammation

Innate immunity is the immunological frontline that defends the host against diverse noxious pathogens. Innate immunity is composed of phagocytosis, intracellular killing antigens, and induction of proinflammatory or antiviral cytokine generation, which is mediated by a variety of cells, including monocytes/macrophages, natural killer (NK) and natural killer T (NKT) cells, dendritic cells (DCs), neutrophils, eosinophils, and basophils in corroboration with amplification of proinflammatory cytokines and other soluble factors. Studies on alteration of the capacity of these cells with aging have established possible mechanisms of innate immune functional disturbance with aging, showing lower expression and impaired pathway of the pattern recognition receptors (PRR) and danger recognition receptors, reduction of intercellular signal, changes in the phagocyte population, and disturbance of phagocyte inducible function [21,22,23,29,30,31,32,33,34,35]. 

Airway epithelial cells are the first interfaces with various environmental pathogens to defend host organs from them in innate immunity. The airway and alveolar respiratory epithelial cells decrease and reduce the mucociliary clearance ability, leading to defects in the clearance of noxious pathogens [18]. The blockade function of airway epithelia has been shown to be disturbed in healthy aged individuals [36]. Fifty-five genes that are involved in barrier function were detected in bronchial brushings from healthy individuals and shown to be significantly associated with aging [37]. Alteration in airway epithelia might allow harmful pathogens to invade the bronchial tissue, leading to increased susceptibility to infectious pathogens.

### 3.1. Change in PRR with Aging

The innate immune pathway initiates the recognition of pathogens via PRR, such as toll-like receptors (TLRs). TLR activation leads to the generation of proinflammatory cytokines through the activation of NF-κB-dependent cascades, production of type-1 interferons (IFNs), and upregulation of INF-dependent genes, resulting in rapid defense against pathogens and subsequent formation of adaptive T and B cell immune responses [21,22,23,29,30,32,33]. The expression of certain TLRs is depressed in immune cells including monocytes, DCs, and leukocytes in aged humans and macrophages in aged mice. TLR1 expression was decreased in aged human monocytes, leading to the reduction of TLR1/TLR2-mediated production of TNF-α and IL-6 [23,38,39,40]. Neutrophils from older individuals also expressed lower TLR1, which may reduce TLR1-related proinflammatory cytokines [30,41]. Levels of TLR2 or TLR4 expression depend on innate immune cells, showing an increase or no change in neutrophils, a decrease in DCs, and no change in macrophages in older humans [23,30,31,32]. The heterogeneity of age-related changes in TLR2 or TLR4 expression on the immune cell surface is likely involved in the sustained magnitude of their mRNA expression in human whole blood leukocytes (PBMC) independent of aging, essentially involving a basic level of TNF-α in older humans [23,30,31,32]. 

The function of downstream signals declines with aging after ignition of TLRs, accompanied with a decrease in those receptors. After stimulation via TLR1, neutrophils from older individuals showed reduced activation marker expression, chemokine production, and signaling intermediate phosphorylation compared to neutrophils from younger individuals [30,39].

The function of older murine TLR2 or TLR4 was also impaired regarding TLR-induced cytokine production [23,30,31,32]. The production of TNF-α by TLR2 or TLR4 agonist-stimulated human PBMC was diminished, associated with aging [23,30,31,32]. In contrast to this, TLR agonist-stimulated monocyte-derived DCs, following activation by IL-4 plus GM-CSF, enhanced cytokine production in older humans [30,39]. Transcriptional profiling revealed that peripheral blood mononuclear cells (PBMCs) stimulated by TLR4, TLR7/8, and RIG-I agonists delayed the downstream signaling pathway, especially in IFN-related signaling in older humans [42]. This evidence suggests that aging in humans decreases the capacity of initial responses to environmental stimulation, leading to reduced elimination of pathogens, impaired and prolonged recovery of tissue injury, and inflammation. Consequently, low-grade inflammation persists in older humans.

### 3.2. Alteration in Neutrophil Function with Advancing Age

Neutrophils serve as crucial effector cells that first respond to microbes or DAMPs to protect hosts. The proper initiation of their response can remit infections, suitably resolve inflammation, and prevent chronic inflammation following persistent nonspecific tissue damage, constituted by proper migration, phagocytosis, intracellular destruction through pH change, destruction by reactive oxygen species (ROS) and cytotoxic enzymes, release of cytotoxic ROS, and degranulation of proteinases [43,44,45]. Neutrophils can also burst network-like structures of chromatin filaments coated with histones and granular and cytosolic proteins, referred to as neutrophil extracellular traps (NETs), to grab and kill pathogens more efficiently [46,47]. 

Studies on neutrophils in aged donors demonstrated a decline of cell properties, particularly regarding innate immunological response [23,30,31,32,41]. Decreased expression of TLR1 in neutrophils in older humans diminished the upregulation of integrin activation markers, such as CD11b and CD18, production of IL-8, and recovery from apoptosis [41]. Neutrophils in healthy older individuals exhibit impairment of chemotaxis, phagocytosis, intracellular erasing, degranulation, generation production of cytokines, and NET [30,31,32,47,48]. Peripheral blood neutrophils were reported to exert less phagocytosis of gram-negative bacteria because of decreased CD16 expression in those cells, which is necessary for Fc-mediated phagocytosis, in healthy older individuals compared to younger individuals [49]. Neutrophils treated with GM-CSF, IL-2, or lipopolysaccharide (LPS) did not remarkably prevent their apoptosis in healthy older individuals compared to younger individuals [50]. This functional deficiency could lead to the insufficient elimination of pathogens in older adults, which could enhance infectious micro-organism colonization of the airways and microbiome population changes [51,52,53,54]. Collaborating with other phagocytes and lymphocytes, impaired neutrophils in older adults could lead to some degree of persistent inflammation, attributed to tissue damage by pathogens.

In contrast to the impaired function of neutrophils, a small study on older patients with mild to moderate asthma reported equal production of LTB4 by GM-CSF-stimulated peripheral blood neutrophils compared to younger individuals [55]. These facts suggest that some specific signaling pathways that initiate the production of proinflammatory mediators are preserved. Neutrophils also maintain an adequate ability to invade from the blood vessels into tissue, including their integrin expression, adhesion to endothelial cells, and migration to inflamed sites [49,56].

Interestingly, a study on the interaction between neutrophils and the inflamed vascular endothelium in aged mice demonstrated that once neutrophils adhered to the vascular endothelium of inflamed tissues, numerous numbers of those attached cells migrated in the reverse direction through the transendothelial layer back to the circulation [57]. Neutrophils that exerted reverse transendothelial migration (rTEM) at the distant inflammatory tissue were also reported to be retained in the lungs of aged mice and to be progressively activated after a while, showing increased expression of CD11b, ICAM-1, neutrophil elastase, CD66α, and CXCR4, and reduced expression of CD62L and CXCR2. The activated appearance of rTEM neutrophils resemble circulating senescent neutrophils in terms of surface expression. Neutrophils morphologically change and express lower CD62L and higher CXCR4, which are considered as senescence, in circulation and migrate back into the bone marrow and are eliminated in daily cycles in mice [58]. Senescent neutrophils were also shown to upregulate several gene pathways, including integrin and leukocyte adhesion, TLR and NOD-like receptor (NLR), and NF-κB signaling pathways, on surface in aged mice, in addition to expressing TLR4, suggesting that neutrophils in the circulation continue to be primed by signals and potentially activated with aging [59]. In aged mice, aged neutrophils highly express an Mb2 integrin under stable conditions and generate more NETs formation in response to LPS, which is a counterpart of TLR4. Additionally, depleted diversity of microbiota reduced senescent neutrophils. Taken together, senescent neutrophils constitutionally receive stimuli in the microbiome milieu and are essentially ready to expand inflammation. These substantial properties might contribute to advancing the pathology in asthma in the elderly because neutrophil counts increase in those patients. Notably, the implications of the evidence obtained on neutrophils in old mice on the pathogenesis of asthma in the elderly needs further study.

### 3.3. Alteration in Eosinophil Function with Advancing Age

The effect of advancing age on eosinophil function is not fully understood. One study reported that IL-5-stimulated blood eosinophils reduced EDN degranulation in older patients with asthma compared to younger patients, while they spontaneously released equal EDN in older patients with asthma between in the older and the younger ones [60]. The equivalent volume of EDN content in older asthma patients compared to the younger suggests that the process of piecemeal degranulation induced by some stimuli was disturbed in the older individuals in that study. However, it is noted that blood eosinophils could be potentially upregulated toward the insufficient level to respond to additional stimulation, particularly IL-5, because these constitutionally predispose to type-2 inflammatory mediators in asthma. This eosinophil condition might resemble exhausted cells, a type of cell senescence, though senescent eosinophils remain not fully understood. Non-specifically stimulated blood eosinophils in older asthmatics also showed diminished production of superoxide anions but exhibited comparable capability of adhesion and migration compared to those in younger individuals. A small ex vivo study on older patients with mild to moderate asthma reported an equal production of LTC4 by GM-CSF-stimulated peripheral blood eosinophils compared to younger individuals [55]. Thus, eosinophils in the elderly patients with asthma seem to conserve most of their functions, except their antipathogen abilities. However, there is no evidence regarding the capacity of releasing tissue injury inducers by eosinophils in response to PAMPs and DAMPs. A study on the comparison between aged and younger adults with asthma regarding EDN and NE level in sputum demonstrated that sputum EDN levels were equivalent between both groups, whereas sputum NE levels were higher in the aged patients with asthma than in the younger, suggesting that eosinophil ability to be degranulated is preserved, whereas that of neutrophils is amplified in the aged asthma patients [61]. Further studies are needed to elucidate the alteration of eosinophils in elderly patients with asthma.

### 3.4. Alteration in DCs Function with Advancing Age

DCs act as novel antigen-presenting cells to connect between innate and adaptive immunity. The functions of DCs are compromised with age in humans, showing restricted capacity to respond to antigens and DAMP through NLR signaling, diminished capacity of antigen phagocyte and presentation, less ability to migrate to lymph nodes, and reduction of proinflammatory cytokines in response to antigens [62,63]. Nevertheless, DCs in aged individuals exhibit enhanced reactivity to self-antigen, such as human DNA and apoptotic cells, through activation of NF-κB, showing increase in IL-6 and IFN-α production by monocyte-derived DCs. Therefore, age-associated changes in DCs might potentially be primed by self-pathogens, leading to the advanced peripheral intolerance and chronic inflammation observed during aging [62].

DCs in aged individuals may affect airway epithelial cells (AECs). DCs from aged individuals chronically produce low magnitude of TNF-α and induce the production of CXCL-10, CCL-20, and CCL-26 by AECs [63] in the ordinary state, attracting T cells, CCR6-expressed AECs, and eosinophils. DCs in aged individuals also produce a disintegrin and metalloproteinase (ADAM), which exhibits degradation of the structural proteins [63]. Age-associated changes in DCs might support the progression of airway inflammation and permeability.

### 3.5. Alteration in NK Cell Function with Advancing Age

NK cells also present age-associated changes on behavior, including reduction of cytokine secretion and decrease in cytotoxicity on target cells such as damaged cells and senescent cells [64]. This alteration is contributed by a change in the NK cell phenotype, for instance, decrease in CD56^+^ bright cells and increase in CD56^+^ dim cells. NK cells are less capable of producing IFN-γ and express perforin, granzyme, and NK cell cytotoxicity with aging [64,65]. Some NK cells with aging showed a reduced ability to eliminate tumor cells or damaged cells, which is attributed to the expression of the inhibitory receptor KLRG1, particularly in CD56^+^ dim NK cells, and both NKG2A and KLRG1 [64]. Reduction of inflammatory mediator production of NK cells promote a defective innate immune response in older humans. Additionally, restriction of eliminating damaged cells preserves the production of inflammatory cytokine and DAMPs and leads to inflammaging.

### 3.6. Alteration in Monocyte/Macrophage Function with Advancing Age

Monocyte- and macrophage-altered magnitude of TLRs expression with aging depend on the TLRs, showing reduced expression of TLR1 and upregulation of *TLR3* mRNA in old human monocytes and no change of TLR2 and TLR4 upregulation in human macrophages [23,30,32,38,39,40,66]. Reduction of TLRs could lead to diminished production TLR-induced cytokines such as IL-6 and TNF-α by monocytes or macrophages in old people [23,30,32,38,39,40].

Along with aging, macrophages in the lungs of mice were reported to decrease and to exhibit downregulation of pathways involved in the cell cycle but upregulation of certain inflammatory pathways, including substance P, PGE_2_, macrophage inhibitory factor, oxidative burst and IL-8, and vascular endothelial growth factor signaling in the transcriptional profile [67]. Lung-resident macrophages impaired phagocytosis and scavenging capacity in old mice [67]. Activated macrophages in the lung were shown to be increased and to poorly respond to further stimulation by IFN-γ in old mice, though those cells released more proinflammatory cytokines in response to mycobacterium tuberculosis infection than those from young mice [68]. Macrophages from old murine lungs reduced production of ATP, accelerated generation of ROS, and decreased antioxidant reflection in response to streptococcus pneumoniae infection [69]. This phenomenon was abolished by improvement of mitochondrial function and reduction of oxidative stress, suggesting pulmonary-resident macrophages impaired mitochondrial function with aging [69].

## 4. Alteration in Adaptive Immunity Associated with Advancing Age

There have been extensive studies on alteration of the adaptive immune system with aging in humans and mice [17,18,19,20,21,22,24,25,26,27,28,70,71,72,73]. As for the initial stage of adaptive immunity, aged DCs have been reported to have an impaired capacity of allergen presentation, affecting the T cell receptor (TCR)-initiated pathway in mice [70]. In addition, most studies on age-related changes in the adaptive immune system have been focused not only on changes of the T cell compartments due to thymus involution, compromised homeostatic proliferation of naïve T cells, and various continuous challenges, but also on subsequent T cell effector dysfunction. Disadvantages in the adaptive immune response are considered as a possible cause and are involved in age-related disease and asthma.

### 4.1. Reduced Activeness of Thymus Affects Peripheral T Cell Population

A decline of thymic activity compromises the T cell compartment, mainly depending on the involution of the thymus [18,26,27]. Thymus involution starts in early life in human subjects, showing that the thymopoietic space is reconstituted with fatty tissue [26,27]. The replacement result in consequences, showing a decrease in greater than 95% of CD4^+^ CD45RO^+^ cells of the thymic output and a 10-fold decrease in TCR diversity in aged humans [73]. However, the naïve T cell compartment is still maintained in humans [26]. The reduction of magnitude of the thymic ability of the peripheral naïve T cell pool is estimated at up to 16% throughout the adult lifetime [26,74].

In aged mice, thymic function rapidly diminishes during adolescence and early adulthood, confirming that new thymic emigrants produced by the thymus of aged mice attenuated the expression of activation markers, their proliferation, TCR function defect, and IL-2 production [20,75].

### 4.2. T Cell Population Paradigm and Function Senescence

The naïve T cell compartment maintains its own size, where the naïve cells homeostatically proliferate [26,27]. Along with aging, homeostatic proliferation is impaired by the following processes: (1) depletion of cell compartment size by loss of fibroblastic reticular cell niches, reduction of *IL7R* gene expression, telomere defect, and cellular senescence; (2) progression of memory cell differentiation by reduced self-renewal and virtual memory T cells, which are cytokine-activated naïve CD8^+^ T cells and are marked in mouse CD8^+^ T cells [26,27]; (3) clonal expansion with some evidence regarding a contraction of the TCR repertoire; and (4) phenotypically central memory T cells infiltration, mainly for CD8^+^ T cells [26,27]. 

A single-cell RNA sequencing and multidimensional protein analysis to characterize CD4^+^ T cells in mice revealed that exhausted, cytotoxic, and activated regulatory T cells were predominant in old mice, showing pro- and anti-inflammatory attributes [76]. In this study, cytotoxic CD4^+^ T cells were predominantly composed of TH1 and TH17 cells and produced IFN-γ, IL-17, and IL-21.

In contrast to CD4^+^ T cells in mice, maintenance of a naïve CD4^+^ T cells compartment receives sufficient benefit from homeostatic proliferation, partly because this process might be supported by IL-2 through expression of CD25 on CD4^+^ T cells and less epigenetic alteration of CD4^+^ T cells with aging in humans [77]. In contrast to the naïve CD4^+^ T cells, the reduction of circulating naïve CD8^+^ T cells is prominent as a consequence of damage of homeostatic proliferation with aging due to epigenetic changes including dysfunction associated with IL-7 [26,77,78,79]. As a result, naïve T cells decline, and therefore, the TCR repertoire results in contraction. Deterioration of TCR diversity with aging is promoted by other factors, including clonal hematopoiesis, independent of antigen specificity, continuous explosion of antigens causing memory inflation, and T cell senescence due to clonal expansion in humans [80]. T effector memory CD45RA cells (TEMRA cells) present mostly loss of TCR diversity, expand under chronic viral infection, such as cytomegalovirus (CMV), sustain effector functionality, enhance inflammatory mediator production, and exhibit aspects of cellular senescence, including cell cycle arrest, expression of DNA damage foci, and production of the senescence-associated secretory phenotype (SASP) [81,82].

The decrease in the naïve T cells population affected by destruction of homeostatic proliferation reflects on the abundant memory T cells, exhibiting the deteriorated feature of memory function. Failure to maintain quiescence engages advanced differentiation pathways in T cell aging, resulting in a terminal differentiated condition. This transition is likely to alter cell performance without cell senescence. A study on microRNA demonstrated that miR-21^hi^-expressed naïve CD4^+^ T cells in old individuals sustained diminishment of several negative regulators of signal pathways and tended to generate inflammatory effector T cells rather than differentiation into memory T cells [83].

There exists a subset of memory CD8^+^ T cells activated by cytokines without experiencing antigen stimulation, named virtual memory T cells. These cells were reported to be accumulated and to present characters like memory T cells in response to exposure to specific antigens in old mice [26,27]. Virtual memory CD8^+^ T cells exhibit high affinity for self-antigens and innate-like effector functions, expressing IL-12R, IL-18R, and IFN-γ, in response to IL-15 in mice; populations of those like virtual memory CD8^+^ T cells in humans accumulate with age and exhibit similar function to those cells in mice [84]. These cells contribute to developing the aspects of cellular senescence in humans and mice [85]. Although differentiation to virtual memory CD8^+^ T cells seems beneficial for innate defense, their loss of proliferative potential results in the disturbance of responses in primary CD8^+^ T cells with aging. 

Taken together, based upon age-related alteration in T cell homeostasis, differentiated subsets of memory T cells compromise innate immune response and act with similar properties as senescent cells, leading to chronicity of infection, dysbiosis, and elevation of proinflammatory cytokines, chemokines, and SASP in old people. Although prompt proinflammatory molecules delivered from subsets of memory T cells support the repair of tissue injury, the violent loop between T cell dysfunction, excessive inflammation, and persistence of stimuli by pathogens promote inflammaging in old people. 

### 4.3. T Cell Senescence

Cell senescence enhances age-associated dysfunction. Senescent cells irreversibly arrest their growth, while DNA damage response is promoted by induction of telomere shortening or continuous response to various stresses [82,86,87,88]. Shortened telomeres are observed in the homeostatic proliferation of naïve T cells and continuous cell replication of memory T cells [26]. Senescent T cells impair their proliferation induced by TCR stimulation and present low telomerase activity and short telomeres and DNA damage signs including γH2AX foci, apoptosis resistance, and β-galactosidase activity [27]. Additionally, senescent T cells fail to express CD27 and CD28 and activate the expression of KLRG1, a terminal differentiation marker [82]. 

Though senescent T cells are involved in the progression of age-related disease, such as cardiovascular, metabolic, and neurodegenerative diseases, the mechanism though which these cells modify the pathophysiology of asthma remain poorly understood.

### 4.4. Contoribution of T Helper 17 (TH17) Cells

An increase in T helper 17 (TH17) cells has been observed in aged humans and aged mice, with more production of IL-17, a representative proinflammatory cytokine derived from TH17 [78,89]. Difference of TH17 cells from naïve T cells is induced by IL-6 and TGF-β, members of SASP. Increment of these cytokines promoted by age-related change in the immune system might be involved in the increase in TH17 cells with aging. Studies on aged individuals and mice demonstrated that regulatory T cells (Tregs) also increased, of which natural Tregs particularly increased, while induced Tregs were reduced [90,91,92]. Additionally, the TH17/Treg imbalance could be advanced in healthy aged persons [89]. The skew toward TH17 cells in healthy aged individuals might enhance neutrophilic inflammation in asthma in the elderly, as observed with severe neutrophilic asthma. In keeping with this, IL-17 and IL-10 were shown to be increased in the older asthma patients compared to the younger [93]. On the other hand, a study on asthmatics demonstrated that IL-17 amplified the production of SASP cytokines by bronchial fibroblasts, including GM-CSF, TNF-α, IL-1β, and IL-6 [94]. These progressive interactions between CD4+ T cells and proinflammatory cytokines might evolve to inflammaging in aged patients with asthma.

### 4.5. Alteration in B Cell Population and Function with Advancing Age

B cells undergo alteration related to senescence [75,95]. In contrast to the maintenance of total T cells percentage, B cells percentage and numbers decrease with aging in humans. Additionally, there is alteration in the distribution of B cell differentiation from naïve to memory B cells with aging. Memory B cells circulating in blood are composed of IgM memory B cells and switched memory B cells. Under B cell senescence, the percentage that switch to memory B cells decrease without change in IgM memory, while the percentage of naive and the subset called double negative (DN) B cells increase. DN B cells are shown as naïve B cells through stimulation of TLRs. Observational studies have demonstrated that DN B cells increase in blood in autoimmune and infectious disease in humans, suggesting that these cells could accumulate in chronic inflammatory disease and are continuously exposed to self-antigens or noxious pathogens [96,97]. Given such evidence, DN B cells are likely a terminal B cell differentiation form with class switch, resulting in the inability to be reactivated by chronic stimuli. Factors other than the alteration in B cells subclass distribution include the decline in the generation of specific protective antibodies and the prolonged survival of plasma cells in older humans [98,99]. The affinity and avidity of antibodies for antigens diminish, likely due to deficiency in the expression of molecules involved in somatic hypermutation (SHM) and Ig class switch recombination (CSR) [100,101] in the germinal center in mice. Evidence on B cell senescence in humans and mice suggests that advancing age compromises B cell differentiation and production of antigen-specific antibodies, leading to impairment of humoral immunity in the elderly.

Memory B cells in aged humans are likely involved in the secretion of SASP, as blood B cells derived from old individuals were reported to highly express *TNF-α* mRNA. These levels were inversely correlated with serum TNF-α and the expression of activation-induced cytidine deaminase, the enzyme that regulates CSR and SHM in B cells [102]. These complicated functions of B cells might contribute to inflammaging.

## 5. Alteration of Lung Parenchyma with Aging

Aging has been associated with lung structural and functional destruction [18,103]. Pulmonary function naturally declines with structural remodeling, including lessened tidal volume and forced expiratory volume in a second (FEV1), and reduction of the gas exchange area and the maximal aerobic capacity [104,105]. Pulmonary tract remodeling is at least in part attributed to the degeneration of the elastic recoil capacity, resulting from reconstruction of the extracellular matrix (ECM) in the lung parenchyma, a decrease in alveolar size, and an increase in acinar airway lumen [104,105,106,107,108]. Age-related pulmonary diseases, such as COPD or idiopathic pulmonary fibrosis (IPF), obviously show lung morphologic alteration with aging. Additionally, aged lungs are vulnerable to environmental agents and infectious antigens, which are affected by lung-residential tissue senescence corroborated with immune senescence.

Various factors promote age-associated lung pathophysiologic modification, including exhaustion and senescence of pulmonary stem cells, renovation of ECM, the altered content of the surfactant system, mucociliary clearance, or the pulmonary nervous system. Studies on the genetic aspects associated with aging revealed characteristics depending on aging [109,110]. RNA sequencing analysis on human bronchial biopsies showed lung senescence signatures, demonstrating that genes that are less expressed with age were involved in the cell cycle regulation, and the damage and repair of the immune system and DNA [109]. Single-cell transcriptome analysis for young and old wild-type mice demonstrated that transcriptomic noise was escalated in most kinds of lung cells with aging, suggesting that epigenetic control diminishes with aging [111]. Further studies are required to establish how various gene aspects are associated with aging-related alterations. In particular, senescent cells or tissue types have a predisposition to cause or contribute to different pulmonary disorders. For example, senescence of bronchial and AECs might be involved in acute or chronic infection and chronic inflammation, that of alveolar epithelial precursor cells might cause IPF, and exhaustion of mesenchymal stem cells might cause COPD [111].

Several studies demonstrated age-dependent airway compromise. The ciliary function of bronchial epithelial cells (BECs) is reported to be reduced in old individuals [112]. Addition to the impaired mucociliary function of BECs, basal and club cells and airway progenitor cells decrease with aging due to less proliferation, reduction in self-renewal, or increase in apoptosis in old mice [18,113,114,115]. Furthermore, in old mice, advancing bronchial wall vulnerability and expansion of inflammatory mediators have also been observed [114]. BEC alteration with aging might disrupt the clearance of noxious pathogens and allow them to invade the bronchial tissue, leading to increased susceptibility to infectious pathogens. In line with this, serum clara-cell protein 16 were reported to increase in old adults with and without asthma, suggesting airway epithelium damage with aging [36]. Epithelial cell adhesion molecule and cadherin1, genes involved in the barrier function of airway epithelia, were detected in bronchial brushings of healthy individuals and were associated with aging [37]. Thymic stromal lymphopoetin (TSLP), a cytokine that is released from bronchial epithelial cells in response to external stimuli and is increased in asthmatics, elevated p21 and p16—both being senescence markers—in human epithelial cells in a dose-dependent manner and promoted remodeling of those cells [116]. Therefore, inflamed or senescent airway epithelia might progress in asthmatics with aging.

Similar to BECs, alveolar type-2 (AT2) epithelial cells showed impaired function due to senescence, including reduction of alveolar epithelial stem cell renewal and differentiation to AT1 cells [115,116,117,118]. Lung-resident stem cells, including AT2 cells, are shown to exhaust and fail to function with aging, contributing to emphysema and pulmonary fibrosis [119].

The transition of mesenchymal compartments is observed with aging. The composition of ECM is modified with aging, showing a decrease in the elastic fiber and collagen XIV and an increase in collagen, especially collagen IV and XVI, in old mice [110,120]. Old human fibroblasts can alter ECM transcription and expression [121]. Single-cell RNA sequencing for old mice found a decrease in collagen XIV expression in aged interstitial fibroblasts accompanied with decolin [110]. Age-related defect of ECM and fibroblast function contributes to the decline in pulmonary function and the pathophysiology of COPD and lung fibrosis.

## 6. Immunosenescence and Inflammaging with Advancing Age

Cumulative immune and other cell activation in response to major stimuli impairs their own differentiation and self-renewal ability due to the defect of the quiescent stage and leads to immunosenescence, demonstrating their exhaustion and senescence, but maintain their productive function despite impairment of defense against pathogens (Figure 1). Aged cells in immunosenescence produce low-grade proinflammatory cytokines, including IL-1β, IL-6, TNF-α, and SASP. These products cross-link with the age-related alterations in lung parenchyma and promote chronic systemic sterile inflammation, termed “Inflammaging”, a concept first proposed by Claudio Franceschi [122].

Evidence on low-grade proinflammatory mediators in older adults and aged patients with disease has demonstrated that chronic low-grade inflammation associated with aging is possibly involved in the pathogenesis of age-related physical alteration and diseases, such as frailty, type-2 diabetes, Alzheimer disease, rheumatoid arthritis, and age-related pulmonary diseases [17,18,20,24,103]. The involvement of inflammaging has been shown in severe asthma and asthma in the elderly [61,62,75,93,123]. Considering the centenarian who has been reported to have increased serum IL-6, not all older adults may experience the disadvantage of inflammaging. It is assumed that a prompt level of inflammatory mediator support is necessary to eliminate excessive inflammatory elements. Further studies are needed to determine whether modification of immune condition in age-related disease results from inflammaging or the principal particularity of the disease with complementary aging factors.

### Initiation of Immunosenescence and Inflammaging

Immunosenescence and inflammaging are engaged and sustained by numerous external and internal stimuli, including allergens, PAMPs derived from pathobionts, fungi and viruses, pollutants, the microbiome, DAMPs derived from impaired cells, abandoned cell debris, and senescent cells (Figure 1). 

The microbiota are recognized as symbiotic communities of various microbes with an equilibrium between the commensal favorable microbes and the potentially harmful ones in various anatomical locations of the body. Along with aging, heterogeneous microbiota, due to increasing pathobionts and opportunistic proinflammatory bacteria, evolves and compromises the immune system, especially in the intestinal tract [51,124]. The cumulative exposure to pathobionts induces the release of inflammatory mediators from chronically activated immune cells, resulting in inflammaging. There have been studies regarding involvement of the microbiota in inflammaging in old mice [125,126,127]. In old mice, the intestinal mucosa develops higher permeability, and gut substances leak into the blood flow, resulting in an increase in bacterial components in the blood [125,126]. Furthermore, circulating monocytes are activated by bacterial components including LPS and produce proinflammatory cytokines such as TNF-α and IL-6 [125]. Circulating neutrophils can be potentially primed by signals and turn into senescent cells in response to the intestinal microbiota in old mice [60].

Notably, constant and systematic low-grade inflammation promoted by dysbiosis, in turn, possibly promote the increase in pathobionts. Similar to dysbiosis, restriction of the lung microbiome diversity has been observed, associated with aging in addition to advanced severity of lung disease, antibiotics administration, and environmental exposure to air pollution and smoking in adults [51,52,53,54]. Studies on airway microbiota in asthma demonstrated the predominance of certain bacteria, and the differences in the lung microbiota among bronchial inflammation types were observed [127,128,129,130,131,132,133,134] and associated with older adults with asthma [128]. In turn, an impaired immune system with aging has also been shown to affect the lung microbiome [53,54].

Oxidative stress has been recognized as an inducible factor of tissue homeostatic dysfunction, such as premature cell senescence and chronic inflammation with aging. Imbalance of oxidation due to excessive ROS production in relation to the antioxidant capacity results in lipid peroxidation, protein carbonylation, and DNA damage within cells with aging [18]. Intracellular ROS is easily produced in response to endogenous and exogenous stimuli, including mitochondrial metabolism, respiratory burst, and exposure to environmental pollutants, such as cigarette smoke, particulate matter, noxious gases, and O3. Excessive amounts of ROS can be released from activated immune cells under oxidative stress during chronic inflammation, leading to the progression of cell damage and further tissue inflammation in an autocrine and paracrine manner. This aggravated process keeps the cells under stress, activating nucleotide-binding domain, leucine-rich repeat-containing family protein-3, and subsequently producing inflammatory cytokines including TNF-α, IL-1β, IL-6, and IL-18, namely SASP [19,20]. Evidence on oxidative stress demonstrated its contribution to the pathogenesis of asthma and age-related pulmonary disorders [18].

While DAMPs are usually stored within cells as endogenous components, they are released from stressed or damaged cells and dead cells in response to environmental pathogens and injury signals [135]. DAMPs consist of ATP, uric acid, high mobility group box 1 (HMGB-1), S100 calcium-binding protein, histones, DNA, and IL-1 family cytokines. DAMPs activate immune cells and lung-resident cells in autocrine and paracrine manners and are involved in the release of proinflammatory cytokines, accumulation of leukocytes, and allergic airway inflammation in mice [136,137,138]. ATP was reported to induce effector functions of human peripheral blood eosinophils and to be involved in eosinophilic asthma [139,140]. Persistent DAMP stress may promote cell senescence and be involved in immunosenescence and inflammaging [17,18]. Studies on aged mice have demonstrated elevation of HMGB-1 and NALP3 inflammasome—which is activated in response to DAMPs—in old mice [141,142]. Additionally, elimination of the NALP3 gene decreased aged-related inflammation in old mice [142].

Mitochondrial dysfunction exists in most tissues and cell types with aging, including incremented mutation and deletions of mtDNA, accelerated oxidation of constructive and functional proteins of mitochondria, changes of lipid composition of mitochondrial membranes, permeabilization of mitochondrial membranes induced by stress, and defects in electron transport chain complexes (ETC) [18]. Change in mTOR/PGC-1 α/β signaling is observed in senescent epithelial cells of the murine lung [143]. Telomere dysfunction, a cause of cell senescence, promotes abrogation of PGC-1α/ERRα transcriptional programs, reduced expression of ETC genes, decreased ATP, and increased ROS in alveolar macrophages, resulting in an extensive innate immune response [144]. Enhanced production of mitochondrial ROS by CD8^+^ T cell has been associated with advancing age in humans [145]. Mitochondrial dysfunction may contribute to inflammaging [146,147].

Cell senescence is a terminal state of replication of cells in which the irreversible restriction of cell division occurs. Studies on cell senescence demonstrated that various aspects of senescence were observed in diverse cells, including lung structural cells and immune cells. Various stimuli generate senescent cells, involving telomere shortening due to exhausted replication, DNA damage, epigenetic modifications, mitochondrial dysfunction, oxidative stress, cytokines, and lost tumor suppressors (Figure 1) [133]. Senescent cells exhibit long survival ability and resistance to apoptosis. Despite replication arrest, senescent cells preserve their metabolic activity and produce various inflammatory agents specified by cell types, named SASPs, including inflammatory cytokines, chemokines, growth factors, matrix metalloproteinase, and DAMPs. SASPs are recognized as a main contributor of inflammaging and can increase inflammation and remodeling in tissues, susceptibility to infection, angiogenesis, and immune cell proliferation, leading to dilation of tissue repair and regeneration. Involvement of SASPs has been observed in older asthmatics and mice using an allergic airway inflammation model [1,61,75,93,123].

## 7. Inflammaging Contributes to Immunopathophysiology in Asthma in the Elderly

Asthma is a chronic allergic inflammation mediated by immune cells and resident airway cells. The underlying comprehensive immune network consists of a complex of type-2-high and -low inflammation in adult asthma. Immunosenescence can modify type-2-high and -low inflammation with SASPs and promote unique inflammaging in elderly patients with asthma, since immune senescence compromises innate and adaptive immunity in aged individuals (Figure 2).

### 7.1. Innate Immunosenescence Can Affect Inflammaging in Elderly Patients with Asthma

Macrophages and neutrophils in aged individuals show a decline in their phagocytotic capability, including increase in susceptibility to pathogens and reduced production of cytotoxic enzymes and radicals [31,32,33]. This age-related impairment delays pathogen elimination and could contribute to the colonization of alternative pathogens and changes in microbiota in the airway, leading to chronic airway inflammation and remodeling in aged adults with and without asthma [127,128,129,131,132]. However, association between airway or gut bacterial dysbiosis and clinical manifestations in asthma is not fully understood. The bacterial diversity is shown to be inconsistent among observational studies, depending on samples, generations of objective, airway inflammatory phenotype, asthma phenotype and severity, and medication [131]. Proteobacteria was reported to be predominant in the airway of asthmatics compared to healthy individuals [131]. Patients with severe asthma, which is frequently observed in older patients [10,11], enriched an Actinobacteria and a Klebsiella genus member [127]. Adult asthma patients who are C. pneumoniae IgG- and IgA-positive exhibited characteristics manifested by the elderly and in severe asthma, showing higher dose of ICS, more fixed airflow, and air trapping [128]. 

Patients with neutrophilic asthma have reduced bacterial diversity and have abundant Proteobacteria phylum [127,129,131]. Age-related alteration in airway bacterial diversity can continuously engage a type-1 immune response with neutrophilic inflammation and complementary eosinophilic inflammation in asthma. Even with enhanced neutrophil dysfunction in aged asthma patients, excessive neutrophil infiltration induces redundant mucus secretion, abundant bronchial damage, and bronchial remodeling, accompanied with eosinophilic inflammation. This noxious loop between infiltrated neutrophils and eosinophils and heterogeneous diversity of bacteria in airways may promote chronic low-grade inflammation and SARP production in aged asthmatics.

Despite little evidence on airway bacterial diversity in type-2 inflammatory asthma, one study demonstrated that an increase in eosinophil counts in the airway might be associated with increases in bacterial taxa, such as Actinobacteria phylum [132], Neisseria, Bacteroides, and Rothia species [130]. Notably, latent Chlamydia pneumoniae and Mycoplasma pneumoniae infections have been reported to contribute to the pathogenesis of asthma, showing an increase in IgE, IL-4, IL-5, IL-8, and IFN-γ in asthma [128]. Increases in IL-13 level and neutrophil ratios in BAL were moderately correlated with the abundance of bacteria in BAL in patients with asthma, suggesting involvement of IL-13 in the remodeling of airways under inflammation due to bacteria [134]. 

Additionally, gut dysbiosis was reported to enhance the allergic airway inflammation in HDM-sensitized old mice with exposure to HDM [148]. Age-related change of the gut microbiome, accompanied with intestinal dysbiosis, could promote higher mucosa permeability and the leakage of bacterial components and SASP into the blood in mice [125,126]. Considering evidence on age-related alteration in airway bacterial diversity in =adults with asthma and aged individuals, in addition to the involvement of gut dysbiosis in inflammation in aged mice, the cumulative exposure to pathobionts induces release of inflammatory mediators from persistently activated immune cells, resulting in unique inflammaging composed of type-1 and -2 inflammations in asthma in the elderly.

#### 7.1.1. Contribution of Neutrophils to Inflammation in Elderly Patients with Asthma

A number of observational studies have demonstrated increased neutrophils in the airway of elderly patients with asthma in diverse conditions, including varied definitions of asthma in the elderly, samples, and measuring methods (Figure 2) [61,62,93,149,150,151,152,153,154,155]. Raito of neutrophils in sputum basically increase with aging in adults with asthma aged ≥ 20 years [155]. Sputum/neutrophil ratios and numbers (approximately doubled) significantly increased in adults with asthma aged ≥ 55 or 60 years compared to younger individuals (20 to 40 years) [61,62,93]. It is noted that the association between increasing neutrophils in the airway and aging could be influenced by the use of ICS in elderly asthmatics because of the reduction of airway eosinophils by ICS. However, an independent correlation between increased airway neutrophils and ICS use was observed even though ICS possesses the ability to reduce eosinophils in the airway [155]. Some patients with severe eosinophilic asthma with a high dose of ICS and/or OCS patients, and certainly severe asthma patients, have frequently exhibited airway neutrophilia. From the perspective of other age-related pulmonary diseases, COPD and IPF can be affected by neutrophilic inflammation in collaboration with the type-1 inflammatory system, including TNF-α, IFNs, IL-6 and IL-8, SARP, and chronically inflamed constitutive lung cell and tissue. Considering the pathogenesis in age-related pulmonary disease with advancing age, constitutional low-grade inflammation with aging could promote the accumulation of neutrophils and lead to the progression of neutrophilic inflammation in the airway of elderly asthmatics.

The rTEM of neutrophils in circulation engages recirculated neutrophils in infiltration to the lung and promotes readiness to response to various stimuli in old mice [59]. Senescent neutrophils also increase and are primed in response to the intestinal microbiota in old mice [60]. Evidence of neutrophil function in old mice supports the expansion of neutrophilic inflammation in elderly asthmatics. In line with this scenario, concentrations of NE in sputum were increased in older asthma patients compared to the younger, the increase in which was correlated with elevation of sputum neutrophil counts in the older but not in the younger [62]. NE elevation in the airway suggests that neutrophilic inflammatory effects are continuously engaged in elderly asthmatics.

#### 7.1.2. Contribution of Eosinophils to Inflammation in Elderly Patients with Asthma

Change of eosinophil functions is reported to vary in elderly asthmatics. An ex vivo study on older adults (55 to 80 years) with asthma demonstrated a reduction of degranulation by type-2 cytokine-stimulated peripheral blood eosinophils and modestly less production of superoxide anions by those that were stimulated by PMA [61]. On the contrary, eosinophils could accumulate at inflammatory sites and produce an inflammatory lipid mediator [56,60]. These results suggest that eosinophils exhibit certain impaired effector functions but persistently accumulate in the tissue of elderly asthmatics. Some studies on the allergic airway inflammation mouse model supports eosinophilic inflammation in elderly asthmatics, showing that ovalbumin (OVA) intra-challenge to older OVA-sensitized mice increased eosinophil counts in lung tissue compared to younger ones [83,152].

In line with ex vivo studies, observational studies demonstrated that eosinophils did not decrease in the airway or blood in elderly asthmatics [38,61,62,93]. Two studies showed that eosinophil counts or ratios in the sputum of elderly asthmatics were comparable to those in young adults with asthma [61,62]. A prospective observational study demonstrated that eosinophil counts in the sputum of asthma patients aged ≥ 60 years were increased compared to those in patients aged 20–40 years [93]. SARP III Cohort demonstrated that, with aging, older adult severe asthmatics showed an increased population of eosinophils, with peripheral blood eosinophil counts of more than 300 cells/μL, which appeared to be similar to that of children [154]. No difference in sputum EDN concentrations was observed between the asthmatics aged ≥60 years and those aged <40 years, the level of which was correlated with sputum eosinophil counts in both the older and the younger patients [62]. These observations suggest that eosinophils, complimentarily upregulated by stimuli other than IL-5 in the airway, could release granule proteins, even though IL-5-treated eosinophils diminished the release of EDN. Combined evidence obtained from the in vivo and ex vivo studies suggest eosinophils might preserve deteriorated but still harmful functionality and initiatively infiltrate into the inflammatory site in elderly patients with asthma.

#### 7.1.3. Enhanced Mixed Granulocytic Inflammation in Elderly Patients with Asthma

While neutrophils and eosinophils are increased in elderly patients with asthma, accumulation of both granulocytes has been observed in them [85], as well as in severe asthma (Figure 2). Cross-sectional studies showed the characteristics of four endotypes based on the distribution of leukocytes in the airway, including the paucigranulocytic, the eosinophil predominant, the neutrophil predominant, and the mixed granulocyte subclass [11,155,156]. The mixed granulocyte subclass and incrementing of both eosinophils and neutrophils in the airways of asthma patients have been reported to be more frequent in severe asthma [156]. Additionally, obvious correlation between eosinophils and neutrophils and between ECP and MPO in airways of asthma patients was observed [157,158]. 

Because of the amplification of neutrophilic inflammation in the airways of elderly patients with asthma, accumulated neutrophilic inflammation could affect allergic inflammation in the airways. In vivo studies demonstrated that peripheral blood neutrophils stimulated by IL-8 or LPS induced the trans-basement membrane migration (TBM) of eosinophils [159,160]. Severe asthmatics also showed increased susceptibility to the TBM of eosinophils by LPS-stimulated neutrophils [160]. Further, severe asthmatics exhibited spontaneous induction of eosinophil TBM, even by unstimulated neutrophils [160]. Taken together, neutrophils, in response to intrinsic or extrinsic stimuli, might promote eosinophil TBM at the inflammation site. In particular, since neutrophils may be potentially primed by various chronic stimuli in severe asthma, they might maintain transmigration to initiated neutrophils in severe asthma. In fact, older adults with asthma more frequently experience severe asthma compared to adolescents and younger adults [11,62]. Increases in both eosinophil and neutrophil counts in sputum were reported in older asthma patients but were not in younger [84]. Additionally, concentrations of sputum EDN were shown to be correlated with both of sputum neutrophil counts and NE in older asthma patients [62].

### 7.2. Adaptive Immunosenescence Can Affect Inflammaging in Elderly Patients with Asthma

As the deteriorated capacity of phagocytes due to aging can promote inflammaging with type-2-high and -low (type-1) inflammations in elderly patients with asthma, the adaptive immunosenescence process observed in elderly individuals could compromise it.

#### 7.2.1. Type-2-High Inflammation in Elderly Patients with Asthma

While evidence on the disadvantage regarding the antipathogen capacity of environmental irritants has emerged, the involvement of immunosenescence in type-2 inflammation is still controversial, especially in allergen-sensitized old mice. The population transition of the T cell differentiation with advancing age and impaired functions in senescent T cells are shown in aged healthy humans or old wild mice. Despite the increase in CD4^+^ effector memory T cells derived from the spleen, these CD4^+^ T cells reduced the expression of Th2 cytokines that were co-cultured with DCs derived from bone marrow and house dust mites (HDM) in older HDM-sensitized mice [161]. However, the older allergen-sensitized mice exposed to allergens showed enhanced expression of IL-4, IL-5, and IL-13 genes and pathologic features of type-2 inflammation in the lung [123,161].

In contrast, a small ex vivo study showed the increased expression of *GATA3* mRNA by anti-CD3/anti-CD28-stimulated PBMCs in elderly patients with asthma [162]. Concentrations of type-2 cytokines were shown to be elevated in elderly patients with asthma [93]. Observational studies on type-2 cytokines in sputum also showed that mean *IL-4*, *IL-5*, and *IL-13* mRNA expression and concentrations of these cytokines were increased in elderly patients with asthma [93,163,164]. Thus, elderly patients with asthma might exhibit a distinct type-2 immunosenescence from that of the aged allergen-sensitized mice, showing that the former leads to the progression of both the systemic and local type-2 immune response, while the latter advances the local type-2 response rather than the systemic type-2 response.

Several studies showed lower levels of serum total IgE in aged asthma patients, while studies using allergen-sensitized older mice showed increases or parity in those levels [123,161]. These observations might be due to B cell senescence, including impairment of IgE production and increases in DN B cells in old humans and decreases in the switched B cells in mice [95,96,97,98,99,100,101,102]. In contrast to total IgE, some allergen-specific IgE antibodies are reported to be elevated in older asthma patients and older allergen-sensitized mice exposed to sensitized allergens [123,153,161].

#### 7.2.2. Type-2-Low (Type-1) Inflammation in Elderly Patients with Asthma

Age-related alteration in lung epithelial cells and in T cells properties and their differentiation compromise innate and adaptive immunity in aged humans, resulting in increased susceptibility to infection and less repair ability. As the adaptive immunosenescence process is observed in aged individuals, detection and replication of CMV DNA in blood has been reported to be highly prevalent in elderly asthma patients compared to younger patients [165]. Since chronic viral infection, such as CMV, increases TEMRA cells and effector memory T cells in older adults, SASP could increase in the elderly patients with asthma. Terminal differentiated T cells and virtual memory CD8^+^ T cells also produce proinflammatory cytokines, such as IFN-γ, under immunosenescence. In line with this evidence, increase in type-1 cytokines, chemokines, and SASP have been observed in sputum and blood of the elderly patients with asthma, accompanied by type-2 cytokines [93]. Mitochondrial dysfunction may also contribute to accelerated activation of immune cells.

#### 7.2.3. Collaboration of Type-1 and -2 Immune System in Elderly Patients with Asthma

As neutrophils can contribute to eosinophilic airway inflammation, type-1 and -2 immunity can collaborate to promote inflammaging in elderly patients with asthma (Figure 2).

IL-17, which serves as a predominant cytokine to promote neutrophil inflammation, promoted the pathogenesis of allergic airway inflammation in an allergic murine model [166,167,168]. IL-17A is capable of amplifying allergic eosinophilic airway inflammation in the presence of IL-13 in wild mice [166]. An older murine allergic model demonstrated that intra-challenge of OVA to older OVA-sensitized mouse increased both type-1 and -2 cytokines, including IFN-γ, IL-1β, IL-6, IL5, and IL-13 [123]. Thus, IL-17 might contribute to the allergic airway inflammation affected by inflammaging in aged mice.

IL-17 progressed the pathogenesis of asthma in severe asthma patients [169,170]. In line with an allergic murine model, IL-17, IL-6, TGF-β—which is included in SASP and initiates the differentiation of naïve T cells to TH17 cells—and IL-10 increase in elderly patients with asthma [93], which is similar to the elevation of the TH17/Treg cells ratio and cytokines derived from these cells in aged people [78]. Thus, evidence on elderly patients with asthma and old mice suggests that TH17 and Treg cells contribute to the pathogenesis of asthma accompanied with inflammaging in elderly patients with asthma. Further studies are needed to better understand the clinical manifestations affected by TH17 and Treg cells in asthma in the elderly.

In line with this, severe asthmatics with neutrophilic subclass in BAL showed elevation of IL-13 in BAL, and those with the mixed granulocytic subclass showed an increase in IL-5 and IL-13 in BAL [93]. The other study on severe asthma demonstrated increases in IL-4 and -5 levels and neutrophil ratios in BAL of severe asthmatics with a high level of IL-13 in BAL compared to those with low of BAL IL-13 and moderate correlation of increases in IL-13 level and neutrophil ratios in BAL with an abundance of potentially pathogenic microbes in BAL of asthmatics [169]. Thus, asthma in the elderly mediates a particular pathogenesis, showing that immunosenesence promotes a unique inflammaging in which type-1 and -2 immunities coexist and collaborate to generate the inflammation.

Asthma with mixed granulocytic infiltration exhibited increased IL-8 and CXCR3 ligands, IP-10, and Mig in sputum and correlations between sum of eosinophils and neutrophils and cytokines and chemokines [171]. CXC chemokines, such as IP-10 and IFNs, are produced by AECs in response to viral argument eosinophile adhesion to counter ligands directly or through enhanced adhesiveness of endothelial cells [172,173]. In line with this, acute asthma exacerbation prolongs airway eosinophilic inflammation after recovering from acute inflammation due to viral infection. This evidence suggests that type-1 cytokines and chemokines can contribute to eosinophilic inflammation in the neutrophilic inflammation milieu in asthma patients with certain conditions. Since older humans are more susceptible to viral infection, type-1 immune cascades may be conveniently upregulated and sustained, leading to complicated inflammaging in elderly asthmatics.

#### 7.2.4. Clinical Impact of Type-1 and -2 Inflammations in Elderly Patients with Asthma

Mixed granulocytic inflammation has been reported to be associated with clinical manifestations in asthma, including increased frequency with age, longer asthma duration, decreased pulmonary function, and condition stability [174,175,176]. The sputum mixed granulocytic subclass has been shown to largely lose interannual pulmonary function compared to the eosinophil predominant, the neutrophil predominant, and the paucigranulocytic subclass in asthma [149,158,177]. Association between mixed granulocytic inflammation and deteriorated transition of pulmonary function in asthma probably matches in elderly patients with asthma because they often have fixed airway obstruction and air trapping in addition to mixed granulocytic inflammation [149,177]. In accordance with several studies showing that eosinophil counts or their large fluctuations are more strongly associated with altered pulmonary function than the neutrophil count in the airway, EDN has been associated with a decline in pulmonary function and NE in elderly patients with asthma [61], suggesting that activated eosinophils contribute to pulmonary function disturbance in these individuals. Patients with mixed granulocytic asthma and the eosinophil dominant subclasses likely experience frequent acute asthma exacerbations [175,177], and older asthma patients with increases in sputum neutrophil counts have more frequent acute exacerbations [93]. Thus, complicated inflammation with eosinophils and neutrophils might promote the clinical pathophysiology of asthma.

Clinical aspects of COPD or the ratio of asthma–COPD overlap are more frequent in older asthma patients [1]. Adding to the amplified decline of fixed airflow due to the mixed granulocytic inflammation, pulmonary function can be modified by lung tract remodeling associated with aging, including decrease in the elastic recoil capacity, a decrease in alveolar size, and an increase in acinar airway lumen [104,105,106,107,108]. This alteration may cause increased frequency of asthma–COPD overlap in older asthma patients.

Considering the evidence on the involvement of mixed granulocytic inflammation in the clinical pathophysiology of asthma in the elderly, both the elevation of type-1 and -2 cytokines and chemokines would contribute to manifestations in these patients. Older patients with asthma with elevation of IL-1β, IL-6, and MIP-3a/CCL20 levels were reported to have a greater risk of hospitalization, along with those with IL-6- and IL-23-deteriorated asthma control [93]. Taken together, type-2-high and type-2-low inflammation and and SASP comprehensively might mediate the pathophysiology of asthma in the elderly.

## 8. Conclusions and Perspectives

This article reviewed the current understanding of the immune pathophysiology regarding elderly patients with asthma. Asthma affected by multifactorial stimuli leads to the progression of its complex immune response throughout the lifespan. Currently, evidence of immune system alteration across all age groups has emerged, and aging serves as a pivotal modifier for the immune system in elderly patients with asthma. Since elderly patients with asthma are likely composed of three clinical subclasses based on their age of asthma onset and duration of disease, including long-standing asthma, late-onset asthma, and recurrent asthma that remitted in childhood, the complexity of their inflammation may have different presentations depending on the level of immunosenescence at the onset age. Unique inflammation in combination with type-2 and non-type-2 immunity results in a composite condition in elderly patients with asthma and accounts for their treatment difficulty. As it is critical to improve the consequences of asthma in older patients, more personalized treatment is needed.

## Figures and Tables

**Figure 1 biomolecules-12-01456-f001:**
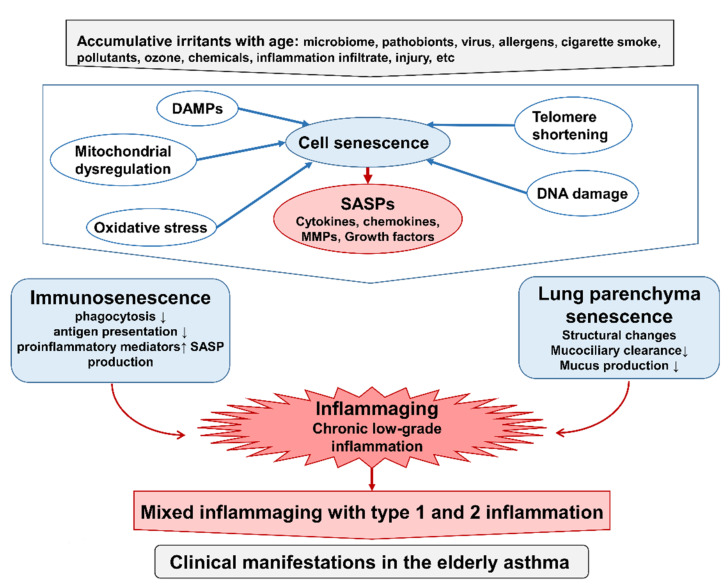
Concept of integrated inflammaging type-2 inflammation in asthma in the elderly. Diverse extrinsic and intrinsic stresses can upregulate various pathways and subsequently activate immune cells and airway epithelial cells or damage these cells. Induction of immune cells senescence, mitochondrial dysregulation, oxidative stress, and damage-associated molecular patterns (DAMPs) in response to these stresses contribute to immunosenescence, accompanied with lung parenchyma senescence. Cell senescence is also initiated by the immune response and produces the senescence-associated secretory phenotypes (SASPs), which lead to immunosenescence progression. Age-related alterations in the immune system promote inflammaging, which is influenced by lung parenchyma aging. Inflammaging may modify type-2 inflammation and promote mixed inflammation with type-1 and -2 immune response in elderly patients with asthma, leading to specific clinical manifestations.

**Figure 2 biomolecules-12-01456-f002:**
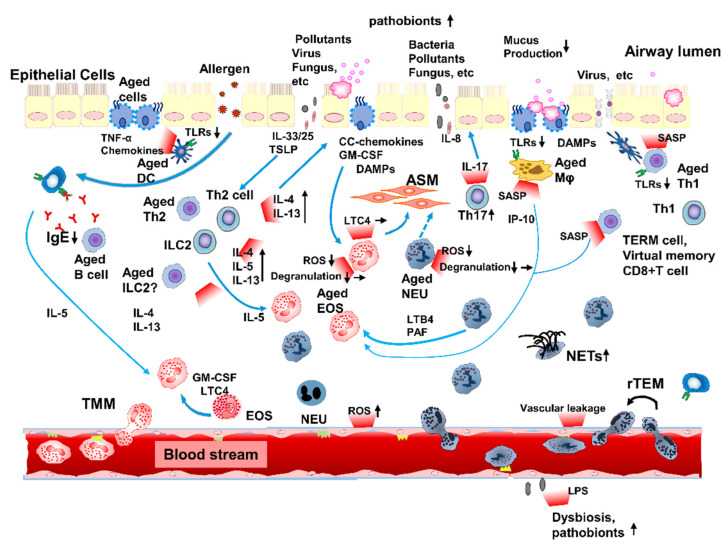
Conspectus of inflammaging in asthma in the elderly. Aged bronchial epithelial cells increase and deteriorate mucociliary function and mucoid production, resulting in increased susceptibility to infection and noxious pathogens. Immunosenescence leads to inflammaging progression accompanied with enhanced type-2 inflammation. Senescent phagocytes play a role by affecting pathogens and influencing the production of proinflammatory cytokines, chemokines, and SASP with the decline of TLRs. Senescent and exhausted T cells generate those mediators and SASP. Type-2 cytokines produced by Th2 cells and ILC2s increase in recognition of IL-33, IL-25, and TSLP, though the effect of aged Th2 cells remains to be elucidated. Th17 cells increase in elderly patients with asthma due to increase in IL-6 and TGF-β. Increase in IL-17 contributes to immune response in circulating neutrophils due to rTEM on distant vascular endothelium, the form of which is modified by dysbiosis, accompanied with mast cells in aged humans. Senescent neutrophils easily promote NETs. Neutrophils activated by IL-8 and LPS through the release of LTB4 and PAF can enhance the accumulation of eosinophils through TMM, which is enhanced in severe asthma. Accumulated eosinophils and neutrophils corporate to promote airway inflammation and remodeling, leading to a stronger decline in pulmonary function. Type-2 inflammation with inflammaging could lead to the progression of unstable asthma conditions.

## Data Availability

Data sharing not applicable.

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
