# Peer review of "Immunosenescence, Inflammaging, and Lung Senescence in Asthma in the Elderly"

_biomolecules, 2022, doi:10.3390/biom12101456_

Round 1

Reviewer 1 Report

Soma and Nagata first reviewed the alterations of immune system through the lifespan.

Then further expanded the review to age related innate immune and adaptive immune response by covering multiple immune cell types, including neutrophil, eosinophil, DCs, NK cells, T cells and B cells. Later, the authors further reviewed the alterations of lung parenchyma with aging, covering healthy and diseased conditions. Eventually, the authors well reviewed immunosenescence and immunoaging effect on asthma of elderly patients.

Overall, the review is thorough and well written. However, one important components of aging related immune response ( immunosenescence) – macrophage was not adequately addressed as other cell types. Thus, this reviewer would suggest the authors considering to expand their coverage on macrophage.

Author Response

Dear reviewer 1, 

We would like to thank you for taking the time to review our work. 

Sincerely, 

Tomoyuki Soma MD, PhD

Reviewer 2 Report

The authors have prepared a review covering an emerging topic in asthma bilogy- immunoaging. In their review they cover immunosenescence and present an overview of age-related immune changes in asthma in elederly. The figures are clear and visually pleasing.

The topics covered are up-to date and interesting to the reader of Biomolecules, however a small section on immunometabolism in eldery asthma would fit even better to the journal.

The paper additionally lacks a concrete structure. For example the authors start with the innate immune system, but they do not cover barrier function in this part. They also put adaptive immunity in the same subchapter (3).

Moreover, the authors continously mix mouse and human data, even though it has long been shown, that mice are not the best translational model for asthma. They for example do not develop it spontanously. So how can the authors be sure and extrapolate data in aged asthmatic mice so easily onto the human population? Therefore mouse and human data should be more clearly separated.

Next, the authors do not report much on the mechanism of these changes. How/when do this 'aged' chages occur? Neutrophils and eosinophils are very short lived cells, is it the environment in the bone marrow? Tissue milliue? There are also big differences between tissue resident and peripheral blood eosinophils (different populations!), this is not explained.

Minor:

line 180 – spelling of neutrophils

line 90, 617- different font

Rewritte line 638-639 – wrong use of english clause syntax.

Line 645 – correlation can not be obvious – it is strong, significant…

Line 657 repeats introduction

Line 756 /757, 774/775… and others- wrong use of english

Author Response

Dear reviewer 2, 

We would like to thank you for taking the time to review our work.

Sincerely,

Tomoyuki Soma MD. PhD

Round 2

Reviewer 2 Report

The manuscript has been sufficiently improved. I have no further comments for the authors.